# Grapevine Xylem Sap Is a Potent Elicitor of Antibiotic Production in *Streptomyces* spp.

**DOI:** 10.3390/antibiotics11050672

**Published:** 2022-05-17

**Authors:** Ramón I. Santamaría, Ana Martínez-Carrasco, Jesús Martín, José R. Tormo, Ignacio Pérez-Victoria, Ignacio González, Olga Genilloud, Fernando Reyes, Margarita Díaz

**Affiliations:** 1Instituto de Biología Funcional y Genómica, Consejo Superior de Investigaciones Científicas, Universidad de Salamanca, C/Zacarías González nº 2, 37007 Salamanca, Spain; ana.mcp@usal.es; 2Fundación MEDINA, Centro de Excelencia en Investigación de Medicamentos Innovadores en Andalucía, Avda. del Conocimiento 34, 18016 Granada, Spain; jesus.martin@medinaandalucia.es (J.M.); ruben.tormo@medinaandalucia.es (J.R.T.); ignacio.perez-victoria@medinaandalucia.es (I.P.-V.); ignacio.gonzalez@medinaandalucia.es (I.G.); olga.genilloud@medinaandalucia.es (O.G.); fernando.reyes@medinaandalucia.es (F.R.)

**Keywords:** *Streptomyces*, elicitor, antibiotic production, xylem sap, grapevine

## Abstract

*Streptomyces* bacteria produce a wide number of antibiotics and antitumor compounds that have attracted the attention of pharmaceutical and biotech companies. In this study, we provide evidence showing that the xylem sap from grapevines has a positive effect on the production of different antibiotics by several *Streptomyces* species, including *S. ambofaciens* ATCC 23877 and *S. argillaceus* ATCC 12596 among others. The production of several already known compounds was induced: actinomycin D, chromomycin A3, fungichromin B, mithramycin A, etc., and four compounds with molecular formulas not included in the Dictionary of Natural Products (DNP v28.2) were also produced. The molecules present in the xylem sap that acts as elicitors were smaller than 3 kDa and soluble in water and insoluble in ether, ethyl acetate, or methanol. A combination of potassium citrate and di-D-fructose dianhydrides (related to levanbiose or inulobiose) seemed to be the main effectors identified from the active fraction. However, the level of induction obtained in the presence of these compounds mix was weaker and delayed with respect to the one got when using the whole xylem sap or the 3 kDa sap fraction, suggesting that another, not identified, elicitor must be also implied in this induction.

## 1. Introduction

*Streptomyces* bacteria are normal inhabitants of soil where they participate in the decomposition of complex carbohydrates, lipids, and proteins by using a wide arsenal of hydrolytic enzymes [1]. These bacteria also produce a wide number of antibiotics and antitumor compounds that constitute their main commercial interest [2,3,4]. The axenic cultivation of different *Streptomyces* strains has been carried out in laboratories for at least 80 years, and the production of secondary metabolites has reached industrial levels. Although some of these *Streptomyces* strains produce up to 4–6 antibiotics under laboratory conditions, genome sequences, obtained from more than 200 species of this genus, have shown that each of them has a wide arsenal of silenced pathways, from 20 to 50 encoding different compounds, that never have been induced under laboratory conditions [5]. Due to the actual health crisis related to multi-resistant bacteria, obtaining new antibiotics is a priority task and, thus, bacteria belonging to this genus comprise a valuable source for the identification of new products.

Several strategies have been employed to induce the expression of these cryptic pathways and some of them have led to successful results. For example: the heterologous expression of these pathways in *S. albus* S4 [6], *S. avermitilis* [7], *S. lividans* [8], or *S. coelicolor* [9,10]; the overexpression of positive regulators in producer strains [11,12,13]; co-culturing with other organisms [14,15,16,17,18,19]; and the addition of elicitors [20,21].

Therefore, the search for new elicitors of this production is one of the strategies to fight antibiotic resistance, leading to improve antibiotic production and the discovery of new bioactive molecules. In this way, the positive effect on the production of different metabolites by adding signaling molecules has been described in different works and summarized in several published reviews [22,23,24,25]. Some of these works refer to the positive effect that the biomass and filtrates of different bacteria and fungi exert over the production of several antibiotics such as natamycin [26,27], valinomycin [28], rimocidin [29], and other interesting compounds. Moreover, a chemogenetic high-throughput screening approach (HiTES) has been developed and successfully used to identify elicitors of silent biosynthetic gene clusters in *Streptomyces* [30]. This methodology identified the antibiotics ivermectin and etoposide as elicitors of the production of 14 new compounds in *S. albus* J1074. These include the antifungal acyl-surugamide A and other molecules that inhibit a cysteine protease implicated in cancer [30].

In this work, we described the elicitor effect that the addition of grapevine xylem sap (*Vitis vinifera* L.) to some laboratory culture media has over the production of antibiotics by 26 strains of *Streptomyces*. We also report the fractionation of the xylem sap components and the identification of some of the elicitor molecules.

Therefore, we demonstrate that the xylem sap from grapevines may be an important source of elicitors to stimulate the production of antibiotics in a wide number of *Streptomyces* species.

## 2. Results

### 2.1. Grapevine Xylem Sap Induces Antibiotic Production in S. coelicolor

One of the goals of our research was to find new elicitors to improve antibiotic production and to discover new bioactive molecules. In this way, the observation of the sap natural leak of grapevines in the spring season of the year led us to explore its natural potential as an antibiotic elicitor. To investigate this possibility, six liters of grapevine xylem sap, collected between April 2018 and April 2019, were mixed to form one homogeneous sample with a pH of 6.

The putative elicitor capacity of the xylem sap was analyzed in the first place over *S. coelicolor* M145 cultured in liquid NMMP, PGA, R2YE, TSB, and YEPD diluted in different proportions with sap. As a control, each medium was diluted in the same proportion using sterile distilled water (Figure 1A). The growth rate was not affected by the addition of the sap in any of the media. 

The qualitative observation of the blue-colored antibiotic actinorhodin (ACT) production after 48–72 h showed a clear induction of ACT in the NMMP and YEPD cultures when diluted one half with xylem sap. No clear effect was observed in the other media used under the same conditions, neither in ACT production in R2YE and TSB, nor in the production of the colored antibiotic undecylprodiginine (RED) in PGA (Figure 1A). Besides, antibiotic activity was clearly detected using the extracts from NNMP and from YEPD cultures grown in presence of xylem sap when assayed against the Gram-positive bacterium *Micrococcus luteus*. By contrast, no antibiotic activity was detected using the extracts from PGA and TSB cultures. Very low antibiotic activity was detected using the extracts from the R2YE control and sap cultures, even though a highly intense blue color was observed under both conditions (Figure 1A). A control with the xylem sap did not show antibiotic activity (Appendix A). The strong stimulatory effect using extracts from liquid NMMP and YEPD media was quantified observing that the production of RED was higher in NMMP than in YEPD while the production of ACT was much higher in YEPD than in NMMP (Figure 1B). 

Moreover, the putative additional metabolites that could be produced in each condition, besides ACT and RED, were also analyzed using liquid chromatography-high resolution mass spectrometry (LC-HRMS) of the NMMP and YEPD culture extracts. In this approach, the overproduction of 8 different compounds was detected (Figure 2 and Appendix A). The compounds overexpressed in NMMP were identified as: deoxydehydrochorismic acid; ε-actinorhodin; γ-actinorhodin; and two unidentified compounds (with molecular formulae of C_15_H_15_NO_6_ and C_32_H_24_O_14_, the latter being a homolog of ACT not appearing in the DNP (Figure 2A). When *S. coelicolor* M145 was grown in YEPD in the presence of sap the compounds overproduced were putatively identified as B 26 or anhydro SEK-4B, SEK-34, SEK-34B, and γ-actinorhodin (Figure 2B). LC-HRMS of the media NMMP and YEPD supplemented with sap demonstrated that none of the detected mass spec signals come from the sap (Appendix A).

Therefore, the molecules present in the xylem sap were able to elicit the production of secondary metabolites efficiently in *S. coelicolor* M145 in some culture media.

### 2.2. Grapevine Xylem Sap Elicits Antibiotic Production in Other Streptomyces Species

The potential stimulatory effect of xylem sap on antibiotic production was also studied in another thirteen additional species of *Streptomyces* (see Section 4) to study if the effect was specific over *S. coelicolor* or not. These thirteen strains, and *S. coelicolor* M145 as the positive control, were grown in liquid YEPD containing either sap or water (control) in a 1:1 ratio for 5 days. 

LC-HRMS detected the presence of several different compounds in the cultures of strains *S. ambofaciens*, *S. argillaceus, S. griseus* subsp. *griseus,*
*S. olivaceus*, and *S. rochei* when grown in the presence of sap (Figure 3, Figure 4 and Figure 5 and Appendix A). However, the cultures of the remaining species, *S. albus*, *S. antibioticus, S. glaucescens*, *S. griseus* IMRU3570, *S. lividans*, *S. parvulus*, *S. steffisburgensis*, and *S. vinaceus* did not present any clear difference regarding the compounds produced by the control and the culture containing xylem sap.

The identification and dereplication of the compounds produced by the different species in the presence of sap were performed by LC-HRMS. *S. ambofaciens* produced one compound identified as nocardamine (Figure 3 left—peak A). *S. griseus* subsp. *griseus* produced chromomycin A3 (Figure 3 right—peak A). *S. argillaceus* produced four new products in the presence of sap (Figure 4 left—peaks A–D): (A) H-indole-5-carboxylic acid; (B) xanthicin; (C) C_13_H_8_N_2_O_3_S, a compound not appearing in the DNP; and (D) mithramycin A. *S. olivaceus* produced six differential peaks in the chromatogram (Figure 4 right—peaks A–F) identified as: (A) antibiotic BA 12100MY1; (B) C_28_H_29_NO_10_, compound with no corresponding compound in the DNP; (C) C_25_H_26_O_10_, five different compounds share this formula (aquayamycin, sakyomicin-A, 2,8-didemethoxy-2′-de-O-methylsteffimycin-D, fridamycin-B, and fridamycin-A); (D) jadomycin Ala; (E) copropophirin III; and (F) 9-C-D-olivosyltetrangulol.

Six different compounds were produced by *S. rochei* in the presence of sap (Figure 5 left—peaks A–F): (A and B) fungichromin; (C) fungichromin B; (D) C_36_H_58_O_13_, a compound not appearing in the DNP; (E) actinomycin G4; (F) actinomycin Xoβ/actinomycin Xoδ; and (G) actinomycin D.

The elicitor effect of the xylem sap was also studied using 12 *Streptomyces* strains from the Fundación MEDINA collection (Appendix A). These strains had previously been grown in 10 different laboratory media and no antibiotic activities had been observed even though the genomes of these strains contain the biosynthetic genes for type I polyketide synthases (PKS I), type II polyketide synthases (PKS II), and non-ribosomal peptide-synthetase (NRPS) (Appendix A). 

When these 12 strains were cultured in YEPD in the presence of sap, the production of new molecules was detected in the strain CA-128791 (Figure 5 right and Appendix A). Three different molecules (peaks A–C) were detected: peak A corresponds to a compound with a molecular formula of C_8_H_6_N_2_O_3_, not included in the DNP; peak B did not generate ions by electrospray and therefore no molecular formula could be assigned; and peak C corresponded to a compound with a molecular formula of C_31_H_37_NO_8_, again, not described in the DNP.

The biological antibiotic activities against *Escherichia coli*, *M. luteus* and *Saccharomyces cerevisiae* were studied for all the *Streptomyces* species producing different compounds (Figure 6). Differential antibiotic activity against *M. luteus* was observed for *S. ambofaciens*, *S. argillaceus*, *S. griseus* subsp. *griseus*, *S. rochei*, and *Streptomyces* CA-128791 when grown in presence of sap. Moreover, *S. rochei* presented also a strong antifungal activity against *S. cerevisiae*. However, *S. olivaceus,* which produced six different molecules, did not have antibiotic or antifungal activities against the microorganisms tested under the conditions used (Figure 6).

### 2.3. Purification of the Elicitor Molecule from Xylem Sap

The grapevine xylem sap was processed as indicated in the Materials and Methods and the capacity to induce antibiotic production of each fraction was analyzed in NMMP liquid cultures (1:1) inoculated with *S. coelicolor* M145. Freeze-dried sap and subsequent solubilization in different solvents demonstrated that antibiotic production inducer molecules were present only in the water-soluble fraction, while no activity was detected in the soluble fractions of ether, ethyl acetate, or methanol. The use of centrifugal filters of different MWCO (molecular weight cut-off) (30, 10, and 3 kDa) to process the water-soluble fraction permitted us to observe that the inducer molecule was in the <3 kDa eluted fraction being active at an equivalent concentration as the whole sap (Appendix A). 

Afterwards, the <3 kDa eluted sample (corresponding to 500 mL of sap) was further fractionated in a Sephadex LH-20 column and 15 samples were obtained (Materials and Methods) and checked for their antibiotic inducer activity. The fraction displaying the highest induction capacity (fraction 6) was analyzed by LC-HRMS and nuclear magnetic resonance (^1^H NMR) spectroscopy. 

The ^1^H NMR spectrum (Appendix A) of fraction 6 from the LH-20 column clearly showed a dominant component displaying only two doublets coupled to each other and resonating at 2.60 and 2.43 ppm. Alongside the dominant component, additional signals in the region between 3.35 and 4.35 ppm could be clearly observed, a frequency range characteristic of hydrogens attached to oxygenated carbons and most likely belonging to a carbohydrate. To identify the chemical nature of the compounds in fraction 6, additional 2D NMR experiments (COSY, HSQC, and HMBC) were acquired (Appendix A). Analysis of such spectra clearly indicated that the main compound in the sample corresponded to citric acid (or its citrate form). The presence of citric acid was corroborated in the LC-HRMS analysis (Appendix A). Likewise, analysis of the 2D NMR spectra confirmed the carbohydrate nature of the second major component. The presence of both oxygenated methylene and methine signals and the absence of any observable anomeric methine signal in the HSQC suggested a carbohydrate of ketose rather than aldose nature. Interestingly, the main component found in the LC-HRMS analysis displayed the molecular formula (MF) C_12_H_20_O_10_ (Appendix A), which is formally equivalent to that of a disaccharide with a formal loss of water; that is, an anhydrodisaccharide. Thus, based on this MF, the ketose nature of the carbohydrate involved, and this specific natural source, we proposed the secondary component in the sample was an anhydrofructosyl disaccharide (Appendix A). Consequently, it is likely that the mentioned anhydrodifructose sugar corresponds to an anhydro form of levanbiose (6-O-β-D-Fructofuranosyl-β-D-Fructofuranose) or inulobiose (1-O-β-D-Fructofuranosyl-D-fructofuranose) (Appendix A). Finally, in the NMR spectra, other minor carbohydrate signals were observed which could correspond to minor di-D-fructose dianhydride isomers or even higher molecular weight fructooligosaccharides (for example, fructose-derived anhydrotrisaccharides) although the presence of this higher molecular weight carbohydrate species could not be corroborated in the LC-HRMS analysis.

To validate these results, our first approach was to make cultures of *S. coelicolor* in NMMP with different concentrations and combinations of potassium citrate, sodium citrate, or/and different amount of fructooligosaccharides (FOS) and to analyze ACT production and antibiotic activity against *M. luteus.* ACT induction occurred when using a mixture of 2 mM of potassium citrate and 60 mM of FOS. However, combining the same amounts of sodium citrate and FOS did not exert induction of antibiotic production. Individually, none of the compounds appeared to have a clear inducer effect on ACT production (Figure 7A and Appendix A). These results indicate a synergy between the potassium citrate and FOS responsible for the elicitor response of antibiotic production. However, the inducer effect of this mixture never reached the same level of induction as that obtained with the sap (Figure 7A). The inducing effect of the 2 mM of potassium citrate and 60 mM of FOS solution was also studied using *S. rochei*, a strain on which the xylem sap induces activities against *M. luteus* and *S. cerevisiae* (Figure 6 and Figure 7B). However, only antifungal activity was detected when 2 mM of potassium citrate and 60 mM of FOS were used as inducers in *S. rochei* cultures (Figure 7B). As no induction of the activity against *M. luteus* was obtained, these results clearly indicate that other inductor component/s, not identified yet, must also be present in the sap.

Since the main identified carbohydrate species in the active fraction correspond to an undetermined di-D-fructose dianhydride related to an anhydro form of inulobiose or levanbiose and since only levanbiose is commercial, different cultures of *S. coelicolor* in NMMP with levanbiose alone or combined with potassium citrate were carried out. No clear induction of the production of ACT was observed when levanbiose was used alone at different concentrations (1, 3, and 5 mM) (Figure 7C). However, clear induction was observed at 5 days of culture when the same amounts of levanbiose were combined with 2 mM of potassium citrate, indicating that the presence of both, potassium citrate and fructose disaccharide, are necessary for inducing antibiotic production (Figure 7C). Again, as the induction was lower with these compounds with respect to the xylem-sap effect, this suggests that other elicitor/s acting synergically with citrate and the di-D-fructose dianhydrides may be present in the original sap and in the equally potent <3 kDa sap fraction. 

## 3. Discussion 

Xylem sap from vines is used in some parts of China as a tonic, for the purpose of preventing aging and to promote health. This fluid is rich in polyphenols, salts, amino acids, organic acids, and carbohydrates and has a highly diverse and complex composition depending on environmental factors and the variety of the vines. Among the mineral elements, the main components are potassium, calcium, magnesium, sodium, phosphate, nitrate, and sulfate. In its composition, nineteen amino acids have also been detected, with Gln, Thr, Glu, and Phe being the most abundant. Organic acids (oxalic, citric, tartaric, and malic) are also present and glucose, fructose, saccharose, and lactose are the main carbohydrates. Nine plant hormones have also been detected, with abscisic acid, auxin, and methyl jasmonate being the most abundant [31,32,33,34,35].

In our study, the xylem sap of grapevines was the source selected to elicit the antibiotic production by *Streptomyces* strains and to characterize this potential by identifying the molecules implied in this process. The results obtained indicate that 27% of the studied *Streptomyces* strains respond positively to this source of elicitors (7 out of 26). Interestingly, one of the strains that responded positively to the xylem sap, strain *Streptomyces* CA128791, had never produced antibiotics in previous studies even using 10 different culture media and knowing that its genome contains genes that encode PKS II (data not shown).

We have demonstrated that potassium citrate and derivatives from fructose are implied in this induction. Therefore, together potassium citrate with FOS or potassium citrate with levanbiose have a positive effect over the production of ACT in *S. coelicolor* and in antifungal activity in *S. rochei*. However, the presence of other unidentified elicitor/s molecule/s in the sap is still possible, since the observed induction with respect to the effect obtained with pure sap or with the <3 kDa fraction does not reach the same level of ACT induction in *S. coelicolor* and does not induce antibiotic activity in *S. rochei*. This synergic activity of several components may also explain the lack of activity of some of the HPLC fractions obtained in a further chromatographic separation of the active LH-20 fraction. The positive effect of potassium citrate (2 mM) has been previously described as an inducer of ACT production when *S. coelicolor* was grown in chemostat culture [36]. However, in our experiments, this salt requires the presence of FOS or levanbiose to induce the production of ACT in *S. coelicolor*, probably due to the different medium (NMMP in our study) and culture conditions used.

The inducer or repressor effect of the carbon source over antibiotics production by *Streptomyces* has been a focus for several research groups and for industrial purposes, and it has been shown that glucose and other carbohydrate sources interfere in the biosynthesis of a wide number of antibiotics. Moreover, studies on the production of nystatin have corroborated that a switch from the utilization of glucose to fructose results in a more than 20-fold increase in *S. noursei* [37]. In fact, enzymes implicated in carbon catabolite repression have been used as targets in mutagenic experiments for strain improvement [38,39]. Through our study, we have detected oligomers of fructose, mainly anhydro forms of fructosyl disaccharides (known as di-D-fructose dianhydrides), as one of the main components of xylem sap. Nevertheless, the addition of different amounts of the related disaccharide levanbiose to *S. coelicolor* cultures in NMMP needs the presence of potassium citrate to exert a positive effect over ACT production.

In conclusion, we have demonstrated that grapevine xylem sap has a potent elicitor effect over the production of a wide number of antibiotics by several *Streptomyces* species, including one strain that had previously been unable to produce antibiotics under laboratory conditions. Additionally, the elicitor action of potassium citrate and FOS, or potassium citrate-levanbiose, has been shown over the ACT production by *S. coelicolor* and antifungal activity of *S. rochei*. However, other minor elicitors that have gone unidentified in this work may be needed to mimic the complete induction observed when using pure xylem sap or the <3 kDa sap fraction.

## 4. Materials and Methods

### 4.1. Strains and Culture Conditions

Bacterial strains *S. albus* J1074, *S ambofaciens* ATCC 23877, *S. antibioticus* ATCC 11891, *S. argillaceus* ATCC 12596, *S. coelicolor* M145, *S. glaucescens*, Tü 49, *S. griseus* IMRU3570, *S. griseus* subsp. *griseus* ATCC 13273, *S. lividans* 1326, *S. olivaceus* Tue22, *S. parvulus* JI2283, *S. rochei* CECT 3329, *S. steffisburgensis* NRRL 3193, and *S. vinaceus*
*JI2838* were obtained from different international collections. In addition, 12 *Streptomyces* sp. strains, isolated from different lichen samples provided by the Fundación MEDINA and preserved in our collection, were also included in this study (Appendix A). Streptomyces strains were grown at 28 °C in NMMP, PGA, R2YE, TSB, or YEPD [40,41,42].

*E. coli* DH5α, *M. luteus* CECT 247, and *S. cerevisiae* W303 2n were used to detect the active compounds produced by the different *Streptomyces* strains. *E. coli* was grown at 37 °C in LB [43] and *M. luteus* and *S. cerevisiae* at 28 °C in YEPD. 

### 4.2. Xylem Sap Collection

Sap was obtained during the spring (April) of 2018 and the spring of 2019 in Salamanca (Spain) by cutting the stems of two 25-year-old grapevines and collecting the liquid droplets under aseptic conditions for 12 days per year. At the end of each day, the liquid collected was frozen. Once the sap had been collected during the springs of 2018 and 2019, all samples were mixed and sterilized together to avoid variability of the contents of the sap collected on different days and years. 

### 4.3. Liquid Cultures and Bioactivity 

*S. coelicolor* M145 was grown in the following liquid media depending on the experiment: NMMP, PGA, R2YE, TSB, and YEPD [41,42]. The control cultures and cultures made with different amounts of sap were set up using 50 mL shaker flasks with three baffles containing 8 mL of the above-mentioned media and inoculated with 5 × 10^5^ spores mL^−1^. For specific induction culture assays, different combinations of potassium or sodium citrate, levanbiose, and FOS were added to the NMMP medium as indicated in the results. The cultures were incubated in an orbital shaker with agitation (200 rpm) at 28 °C for 3–6 days. The cultures of the remaining strains of *Streptomyces* used in this work were mainly grown in YEPD, as indicated in the Results section.

Colorimetric quantification of RED and ACT production from *S. coelicolor* was determined using the previously described spectrophotometric method [44]. All experiments were performed at least in triplicate. 

Antibiotic and antifungal bioactivities were assayed by extracting 1 mL of each culture with 700 µL of ethyl acetate acidified with 1% formic acid and by drying the organic layer in a speedvac. The dry extracts were resuspended in 500 µL of dimethyl sulfoxide (DMSO) and different amounts of the resulting samples were assayed. The extracted samples were then deposited in a hole made using a cork borer (0.7 cm Ø) in either LB plates inoculated with a lawn of *E. coli* or in YEPD plates inoculated with a lawn of *M. luteus* or *S. cerevisiae.* Inhibitory activity was monitored after the plates were incubated at 37 or 28 °C for 1 or 2 days depending on the organism used in the assay. In all the experiments, the same amount of DMSO was used as negative control given no activity. The experiments were performed in triplicate.

### 4.4. LC-HRMS-Analyses

Samples of 1 mL of the different cultures were extracted using acidified ethyl acetate (ethyl acetate containing 1% formic acid) and were dried and resuspended in dimethyl sulfoxide:methanol (1:1) or in methanol for subsequent LC-HRMS analysis of the molecules produced. The analyses were performed as previously described using a Bruker maXis QTOF mass spectrometer coupled to an Agilent 1200 LC [45,46]. Differential peaks were selected by direct comparison of the UV chromatograms and were identified according to the Dictionary of Natural Products (DNP v28.2) available at http://dnp.chemnetbase.com/intro/index.jsp (accessed on 25 April 2022).

### 4.5. Xylem Sap Fractionation and Elicitor Purification and Identification

To characterize the xylem sap molecules able to induce antibiotic production, three 30 mL samples of grapevine xylem sap were lyophilized. The dried matter was extracted using 10 mL of an organic solvent, such as ether, ethyl acetate, or methanol, by shaking for 30 min at room temperature. The insoluble fractions of these samples were again dried and resuspended in 10 mL of water. The water-soluble samples were fractionated using centrifugal filters of three different MWCO (30, 10, and 3 kDa). The capacity to induce antibiotic production of each fraction, organic and water-soluble fractions, was analyzed by adding them to liquid cultures of *S. coelicolor* in NMMP in an equivalent concentration as in the whole sap.

For elicitor purification, two lyophilized tubes containing the sap fraction smaller to 3 kDa, obtained from 250 mL of grapevine xylem sap, were extracted with 50 mL of methanol by shaking for 30 min. After centrifugation at 5000× *g* for 10 min, the insoluble fraction was again dried and resuspended in 25 mL of water. This soluble fraction was processed using centrifugal filters of three different MWCO (30, 10, and 3 kDa). The fraction eluted using the 3 kDa filter was run in a Sephadex LH-20 column with a gravity flow of 1.25 mL/min in a 200 mm × 22 mm column, with 100% water for 150 min. Fifteen 10 mL samples were collected using UV detection at 210 nm. All the fractions were lyophilized and conserved at −20 °C.

Compounds present in the active fraction were analyzed by LC-HRMS as previously reported using an analytical Atlantis T3 column, 20 min run [45,46]. Mass spectrometry dereplication was performed by comparing the retention time and exact mass of the detected components against the high-resolution mass spectrometry database made available by the *Fundación* MEDINA (Granada, Spain). For the components with no matches in the MEDINA database, the obtained molecular formula/exact mass was searched against the Chapman and Hall Dictionary of Natural Products. To carry out the NMR analyses, a dried sample of the active fraction was dissolved in D_2_O and transferred to a 1.7 mm tube. Acquisitions were carried out on a Bruker AVANCE III 500 MHz spectrometer equipped with a 1.7 mm TCI micro-cryoprobe. All spectra were registered at 24 °C.

## Figures and Tables

**Figure 1 antibiotics-11-00672-f001:**
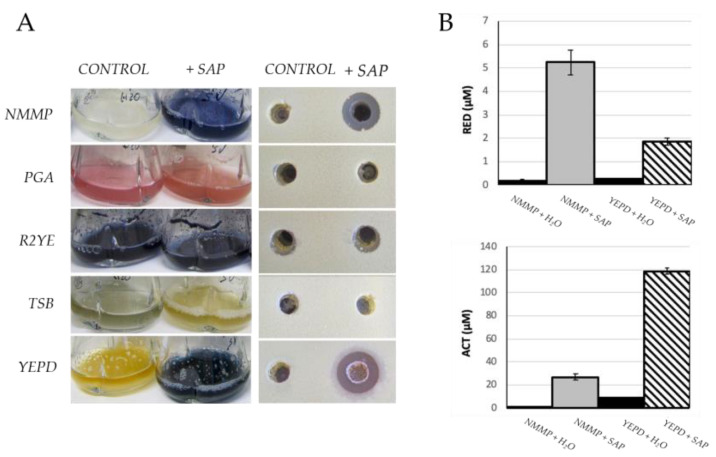
(**A**) Effect of grapevine sap over the production of antibiotics by *S. coelicolor* grown in the indicated medium (NMMP, PGA, R2YE, TSB, and YEPD) diluted by one-half with water (CONTROL) or with sap (+SAP). Left panel: qualitative observation of colored antibiotic production of the different cultures (blue: ACT; red: RED); right panel: antibiotic activity against *M. luteus*, 100 μL of the corresponding extract (CONTROL or +SAP) were deposited in each well. (**B**) Quantification of RED and ACT produced by *S. coelicolor* grown in NMMP or YEPD media diluted by one-half with water or grapevine xylem sap. All the experiments were performed four times with three biological replicates each.

**Figure 2 antibiotics-11-00672-f002:**
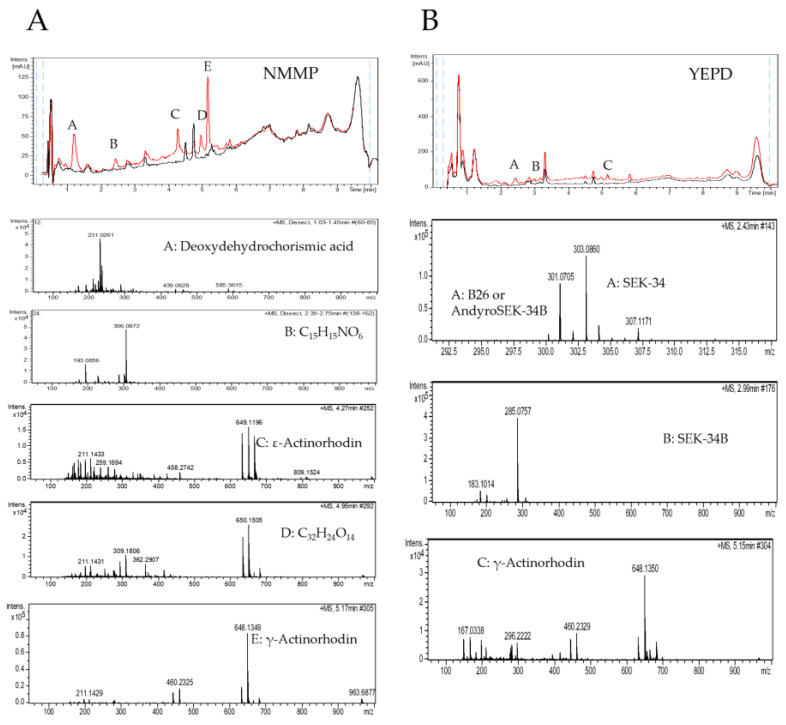
Analysis of the metabolites produced by *S. coelicolor* M145 grown in NMMP (**A**) or in YEPD (**B**) with water (black) or with SAP (red) and detected by UV/Vis absorbance (200–900 nm). Mass spectra of compounds in the differential peaks.

**Figure 3 antibiotics-11-00672-f003:**
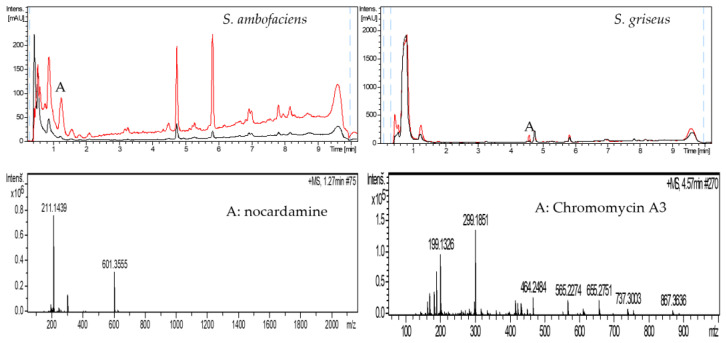
Analysis of the metabolites produced by *S. ambofaciens* (**left**) and by *S. griseus* subsp. *griseus* (**right**) grown in YEPD with water (black) or with SAP (red). Mass spectra of compounds in the differential peaks.

**Figure 4 antibiotics-11-00672-f004:**
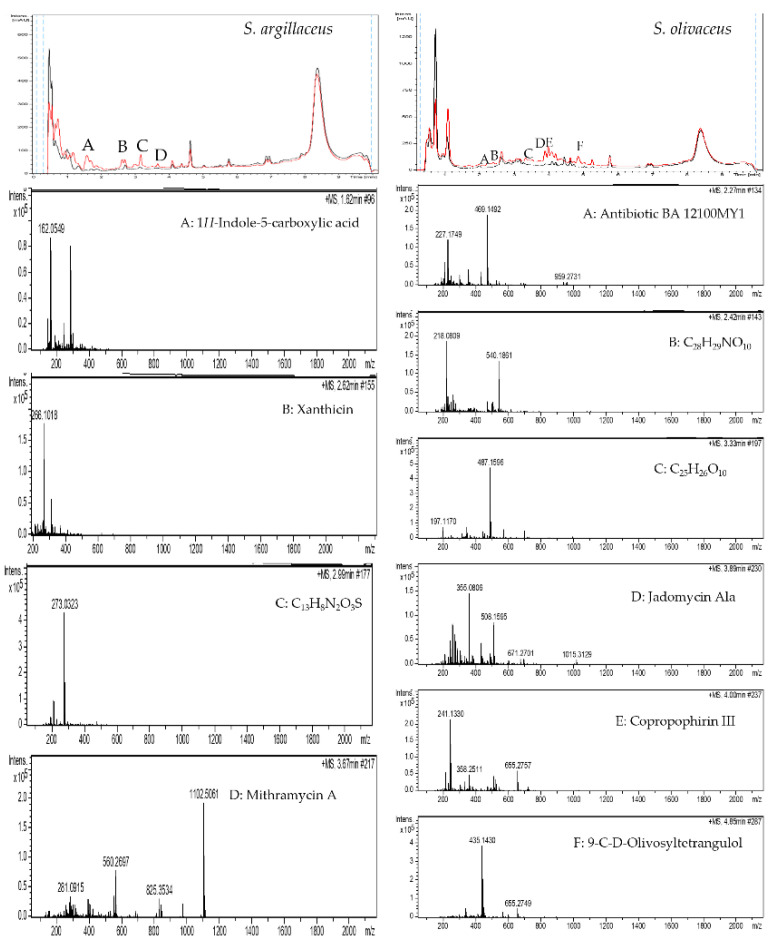
Analysis of the metabolites produced by *S. argillaceus* (**left**) and by *S. olivaceus* (**right**) grown in YEPD with water (black) or with SAP (red). Mass spectra of compounds in the differential peaks.

**Figure 5 antibiotics-11-00672-f005:**
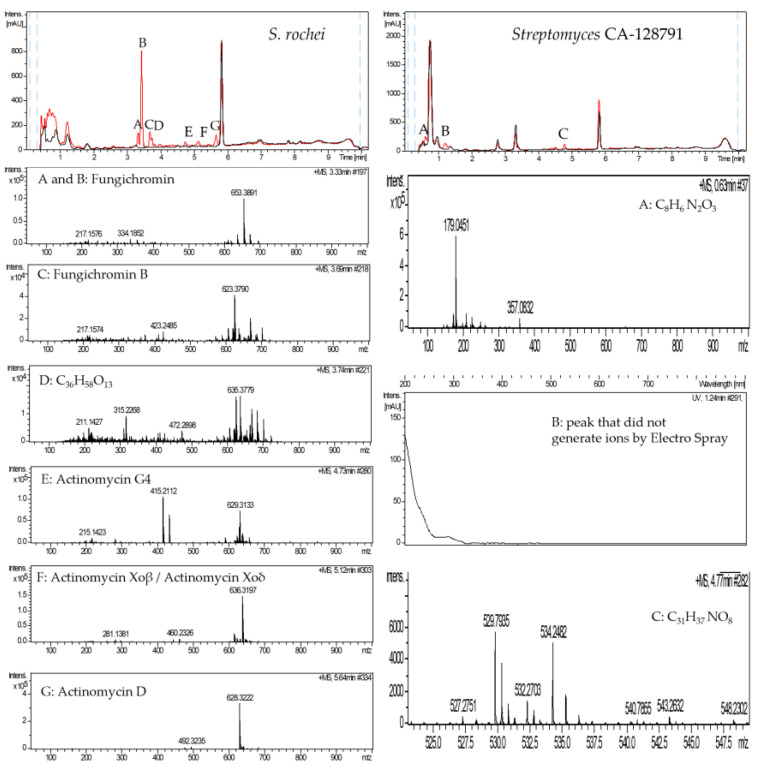
Analysis of the metabolites produced by *S. rochei* (**left**) and by *Streptomyces* CA-128791 (**right**) grown in YEPD with water (black) or with SAP (red) detected by UV/Vis absorbance (200–900 nm). Mass spectra of compounds in the differential peaks.

**Figure 6 antibiotics-11-00672-f006:**
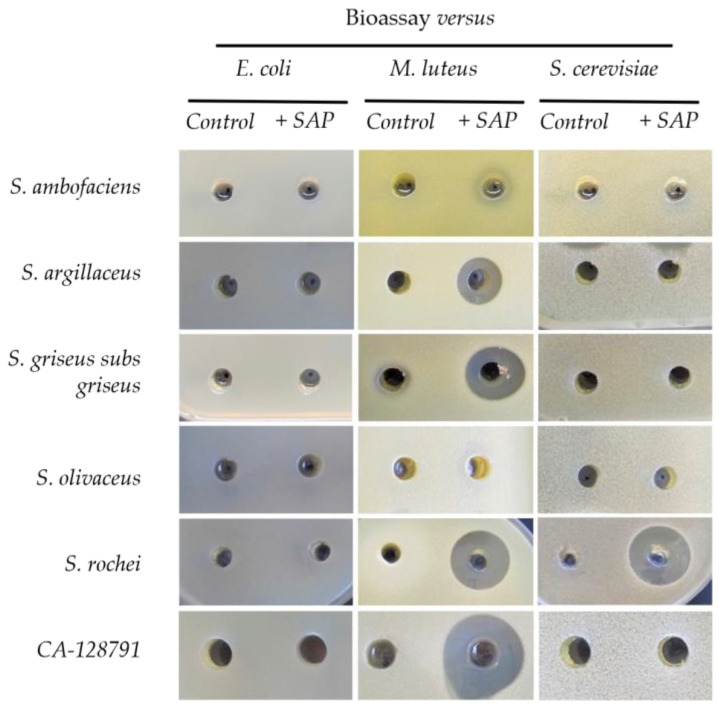
Bioactivity of 100 μL of the extracts of the indicated *Streptomyces* strains in the absence (Control) or the presence of sap (+SAP) against *E. coli*, *M. luteus*, and *S. cerevisiae*. All the experiments were performed four times with three biological replicates each.

**Figure 7 antibiotics-11-00672-f007:**
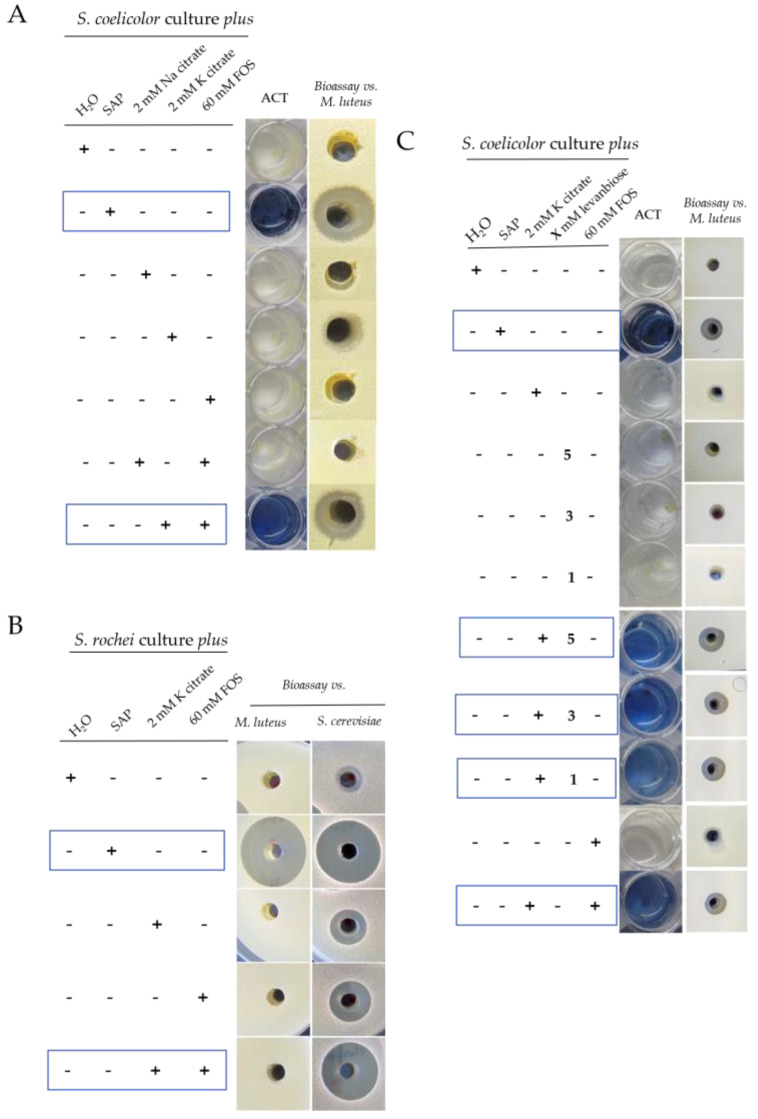
(**A**) Left, ACT production (blue color) by *S. coelicolor* induced by different combinations of potassium or sodium citrates and FOS in NMMP Medium. Right, bioactivity of 100 µL of the extracts against *M. luteus* of the indicated samples. (**B**) Antibiotic and antifungal production by *S. rochei* induced by different combinations of potassium citrate and FOS in NMMP medium. (**C**) Left, ACT production by *S. coelicolor* induced by different combinations of potassium citrate, levanbiose, and FOS in the NMMP. Right, antibiotic activity against *M. luteus* of the indicated samples. Negative control: NMMP + H_2_O; positive control: NMMP + SAP.

## Data Availability

Not applicable.

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
