# Peer review of "Grapevine Xylem Sap Is a Potent Elicitor of Antibiotic Production in Streptomyces spp."

_antibiotics, 2022, doi:10.3390/antibiotics11050672_

Round 1

Reviewer 1 Report

The study demonstrated grapevine sap may be an elicitor for antibiotic production for Streptomyces.

I have below comments:

  1. In general, please improve the quality of the figures, details are difficult to read.
  2. For all antibiotic activity assays, a control of SAP only needs to be added. To be more specific, researchers need to demonstrate the antibiotic activities did not come from SAP itself.
  3. For all mass spec analysis of compounds elicited by SAP (figure 4 and 5), a control of SAP only should be added to demonstrate the detected mass spec signals did not come from SAP.
  4. Please provide a table HRMS lists of all compounds with expected monoisotopic mass value and its observed value. Figure 4 and 5 are slightly difficult to read for this information.
  5. What is the reason that a combination of FOS and potassium citrate can elicit antibiotic production? Any comments or speculation? 
  6. Do any of the tested strains live in the environment which may have access to grapevine xylem sap? If not, testing strains with co-existing conditions may be interesting.

Author Response

  1. In general, please improve the quality of the figures, details are difficult to read.

The figures have been improved.

2.-For all antibiotic activity assays, a control of SAP only needs to be added. To be more specific, researchers need to demonstrate the antibiotic activities did not come from SAP itself.

This control has been added as a Supplementary figure (Fig. S1)

3.- For all mass spec analysis of compounds elicited by SAP (figure 4 and 5), a control of SAP only should be added to demonstrate the detected mass spec signals did not come from SAP.

 A supplementary figure has been added (Fig. S2)

4.- Please provide a table HRMS lists of all compounds with expected monoisotopic mass value and its observed value. Figure 4 and 5 are slightly difficult to read for this information.

  A supplementary table (Table S1) has been added with all the data corresponding to figures 2, 3, 4 and 5

5.- What is the reason that a combination of FOS and potassium citrate can elicit antibiotic production? Any comments or speculation? 

As stated in the manuscript, the effect of potassium citrate alone over actinorhodin production has been described previously. Also, the effect of fructooligosacharides in bacteriocin production has been described for several strains of Lactobacillus (Muñoz et al. Anaerobe 2012,18(3):321-30). Apparently, in our study there is a synergistic effect of both compounds, although, currently, we do not have any evidence that could explain their mode of action.

6.- Do any of the tested strains live in the environment which may have access to grapevine xylem sap? If not, testing strains with co-existing conditions may be interesting.  

 As far as we know none of the strains studied have been isolated from grapevine rhizosphere. For future work, we will harvest new xylem sap having in mind to centrifuge the sap and inoculate plates with the precipitate looking for Streptomyces strains that can be endophytes.

Reviewer 2 Report

Overall, the manuscript is technically sound and the research ideas appear justified. However, the resolution of the figure is very low quality.

Author Response

1.- Overall, the manuscript is technically sound and the research ideas appear justified. However, the resolution of the figure is very low quality.

The figures have been improved

Reviewer 3 Report

The manuscript entitled „Grapevine Xylem Sap is a Potent Elicitor of Antibiotic Production in Streptomyces spp.” describes an interesting study aiming at the development of new antibiotics induced by grapevine xylem sap.

The manuscript is well written and easy to follow. The idea is very valid and needed, and the study design is well planned.

I have only some editorial comments:

The introduction is very well written, logical and perfect till line 62. Then, there are elements which should not be placed in the introduction. Why did the authors explain their results in the introduction? This part should present only what was done by others, not a brief summary of this work. What is the point of reading further if all that is important is given here? At the same time, the intro should contain a hypothesis and the aim of the study, which are not existing here. Please change it.

Please explain all the abbreviations in the first use. Also the ones in line 101. Please explain in actual first use, not in the material and methods.

Why do the captions of the figures have different fonts? The captions should not be merged with figures. Please remove it and add the captions according to the journal requirements.

Line 270: please provide the references to these previous studies.

Author Response

1.- The introduction is very well written, logical and perfect till line 62. Then, there are elements which should not be placed in the introduction. Why did the authors explain their results in the introduction? This part should present only what was done by others, not a brief summary of this work. What is the point of reading further if all that is important is given here? At the same time, the intro should contain a hypothesis and the aim of the study, which are not existing here. Please change it.

All that part has been deleted and summarized in the following paragraph:

In this work we described the elicitor effect that the addition of grapevine xylem sap (Vitis vinifera L.) to some laboratory culture media has over the production of antibiotics by 26 strains of Streptomyces.  We also report the fractionation of the xylem sap components and the identification of some of the elicitor molecules.

2.- Please explain all the abbreviations in the first use. Also the ones in line 101. Please explain in actual first use, not in the material and methods.

This has been corrected

3.- Why do the captions of the figures have different fonts? The captions should not be merged with figures. Please remove it and add the captions according to the journal requirements.

This has been corrected

4.- Line 270: please provide the references to these previous studies.

 These are not published results from the “Fundación Medina” coauthors. The sentence “data not shown” has been added to the text.  

Reviewer 4 Report

In this study, the authors explored the effects of using Grapevine Xylem Sap as the elicitor to induce the biosynthesis of antibiotics in Streptomyces strains. Successful biosynthesis of some known and unknown antibiotics was observed after the induction by the sap. Thus, the major components in the sap were identified and purified. The re-introduction of these compounds/components validated the effect of elicitor, though not as efficient as the original sap. The paper is well-written and well-organized. I therefore recommend the publication of this paper. Below are my comments that can be referred to improve the paper:

  1. My only major concern is regarding the section 2.3. While the authors can identify the main components in the sap, the re-introduction of these compounds (Figure 7A) was less effective in activating the antibiotic biosynthesis. This was likely due to a sub-optimal concentration ratio in the manually prepared solutions. Thus, I would recommend the authors provide the data to show how the alterations of the component concentration would affect the biosynthesis of antibiotics. This would be beneficial for determining the main elicitor(s) and also provide more valuable insights in the field.
  2. Line 24, the font of the “di-D-fructose dianhydrides” needs to be adjusted in accordance with other words in the abstract
  3. Line 67-71, please add a table to list the detailed information of the strains used in this study (and call out when needed in the manuscript). It is not necessary to list all the strains here.
  4. Line 146-153, please check the format here.
  5. Some contents in the figure are not easy to see and read, please provide high-resolution figures when submitting the revised paper.

Author Response

1.- My only major concern is regarding the section 2.3. While the authors can identify the main components in the sap, the re-introduction of these compounds (Figure 7A) was less effective in activating the antibiotic biosynthesis. This was likely due to a sub-optimal concentration ratio in the manually prepared solutions. Thus, I would recommend the authors provide the data to show how the alterations of the component concentration would affect the biosynthesis of antibiotics. This would be beneficial for determining the main elicitor(s) and also provide more valuable insights in the field. 

 A supplementary figure (Supplementary Fig.S10) has been added

2.- Line 24, the font of the “di-D-fructose dianhydrides” needs to be adjusted in accordance with other words in the abstract,

The font has been corrected. It is a mistake of the document done by the journal software. In our original Word file it has the same size than the rest of the document.

3.- Line 67-71, please add a table to list the detailed information of the strains used in this study (and call out when needed in the manuscript). It is not necessary to list all the strains here.

Attending to the comments of reviewer 3 this part has been shortened and now reads:

In this work we described the elicitor effect that the addition of grapevine xylem sap (Vitis vinifera L.) to some laboratory culture media has over the production of antibiotics by 26 strains of Streptomyces.  We also report the fractionation of the xylem sap components and the identification of some of the elicitor molecules.

4.- Line 146-153, please check the format here.

 This is a format problem in the Word file generated by the Journal software. In the original manuscript all the document has the same margins.

5.- Some contents in the figure are not easy to see and read, please provide high-resolution figures when submitting the revised paper.

The figures have been improved

Reviewer 5 Report

This manuscript entitled "Grapevine Xylem Sap is a Potent Elicitor of Antibiotic Production in Streptomyces spp.” described an interesting exploration on the xylem sap from grapevines that as the elicitor to produce different antibiotics from several Streptomyces species. Firstly, the authors established the approach on the grapevine xylem sap inducing antibiotic production in Streptomyces coelicolor with ACT and RED as the indicators. Then, expanding such strategies into other 13 different strains accessed and resulted in the discovery of four unknown compounds, besides discovered known important compounds including actinomycin D, chromomycin A3, fungichromin B, mithramycin A, etc. Finally, the author screened that the combination of potassium citrate and di-D-fructose dianhydrides seemed to be the potential inducers. Although the experiments were well designed and performed, the data and conclusions presented here were a little bit preliminary.

Suggestions and comments for improvement are as follows:

  1. What is the reason for the selection of the grapevine xylem sap as the elicitor to induce the production of antibiotics in Streptomyces.
  2. The structure determination of the potential new compounds should be carried out by the analysis of the 1D- and 2D-NMR.
  3. What is the concentration of the elicitor molecule in the grapevine xylem sap that used as the elicitor.
  4. Which species of grapevine is selected and why selected the 25-years-old grapevines.
  5. The resolution of all the HPLC profiles is too low to obtain the completed information.

Author Response

1.- What is the reason for the selection of the grapevine xylem sap as the elicitor to induce the production of antibiotics in Streptomyces.

Harvesting of xylem sap from grapevines is very easy in the spring. If these plants are pruned when they begin to sprout in spring, the sap comes out of the cut in drops continuously for several days. This does not happen with other plants where we live.

2.- The structure determination of the potential new compounds should be carried out by the analysis of the 1D- and 2D-NMR.

We agree with the reviewer in the interest of the determination of the structure of the new compounds and that will be our future research in this field.

3.- What is the concentration of the elicitor molecule in the grapevine xylem sap that used as the elicitor.

We have not measured the concentration of these compounds in our xylen sap.

4.- Which species of grapevine is selected and why selected the 25-years-old grapevines.

These plants are from RIS grapeyard, one of them is Palomino and the other Verdejo.

5.- The resolution of all the HPLC profiles is too low to obtain the completed information.

The figures have been improved.

Round 2

Reviewer 1 Report

I am satisfied with the comments to my questions provided by the authors, thanks.

Reviewer 5 Report

The authors had revised accordingly, and I wolud like to recommend publication in this journal.